# REST: a preoperative tailored sleep intervention for patients undergoing total knee replacement – feasibility study for a randomised controlled trial

Wendy Bertram [1,2] Chris Penfold [1] Joel Glynn,[3] Emma Johnson,[3] Amanda Burston,[1] Dane Rayment,[4] Nicholas Howells,[4] Simon White,[5] Vikki Wylde [1,2] Rachael Gooberman-Hill [2,3] Ashley Blom,[6] Katie Whale [1,2]

¹Musculoskeletal Research Unit, University of Bristol Medical School, Bristol, UK
²NIHR Bristol Biomedical Research Centre, Bristol, UK
³University of Bristol Medical School, Bristol, UK
⁴North Bristol NHS Trust, Westbury on Trym, UK
⁵Cardiff and Vale University Health Board, Cardiff, UK
⁶The University of Sheffield, Sheffield, UK

**Correspondence to**
Wendy Bertram;
wendy.bertram@bristol.ac.uk

## ABSTRACT

**Objectives** To test the feasibility of a randomised controlled trial (RCT) of a novel preoperative tailored sleep intervention for patients undergoing total knee replacement.

**Design** Feasibility two-arm two-centre RCT using 1:1 randomisation with an embedded qualitative study.

**Setting** Two National Health Service (NHS) secondary care hospitals in England and Wales.

**Participants** Preoperative adult patients identified from total knee replacement waiting lists with disturbed sleep, defined as a score of 0–28 on the Sleep Condition Indicator questionnaire.

**Intervention** The REST intervention is a preoperative tailored sleep assessment and behavioural intervention package delivered by an Extended Scope Practitioner (ESP), with a follow-up phone call 4 weeks postintervention. All participants received usual care as provided by the participating NHS hospitals.

**Outcome measures** The primary aim was to assess the feasibility of conducting a full trial. Patient-reported outcomes were assessed at baseline, 1-week presurgery, and 3 months postsurgery. Data collected to determine feasibility included the number of eligible patients, recruitment rates and intervention adherence. Qualitative work explored the acceptability of the study processes and intervention delivery through interviews with ESPs and patients.

**Results** Screening packs were posted to 378 patients and 57 patients were randomised. Of those randomised, 20 had surgery within the study timelines. An appointment was attended by 25/28 (89%) of participants randomised to the intervention. Follow-up outcomes measures were completed by 40/57 (70%) of participants presurgery and 15/57 (26%) postsurgery. Where outcome measures were completed, data completion rates were 80% or higher for outcomes at all time points, apart from the painDETECT: 86% complete at baseline, 72% at presurgery and 67% postsurgery. Interviews indicated that most participants found the study processes and intervention acceptable.

**Conclusions** This feasibility study has demonstrated that with some amendments to processes and design, an RCT to evaluate the clinical and cost-effectiveness of the REST intervention is feasible.

**Trial registration number** ISRCTN14233189.

## STRENGTHS AND LIMITATIONS OF THIS STUDY

⇒ COVID-19 restrictions in place during delivery required study procedures to be redesigned to enable remote data collection.

⇒ Data collection at the 3-month postsurgery time point was limited due to the volume of operations performed during the study.

⇒ This study was undertaken in two National Health Services hospitals, which demonstrates that it is feasible to undertake a full trial in these settings, however, the findings may not necessarily be generalisable to other settings.

⇒ Embedded qualitative work provided important insight into final study design to support acceptability and participant engagement with a full trial.

## BACKGROUND

Over 100 000 total knee replacements (TKRs) are performed yearly in the UK.[1 2] The primary reason for surgery is severe chronic pain and functional limitation due to end-stage osteoarthritis. The aim of TKR is long-term pain relief and improved function. Outcomes after knee replacement surgery are good, and surgical complications are rare, however, approximately 20% of patients report dissatisfaction due to ongoing pain and functional limitations.[3 4]

Sleep issues are a substantial problem for people awaiting joint replacement; 60%–75% of people with osteoarthritis and 70% of patients awaiting joint replacement experience sleep problems, which increase with condition severity.[5–8] Patients report issues with sleep onset and maintenance, and middle of the night waking.[9]

Poor sleep causes worsening joint pain, depressive symptoms, lower physical activity and increased risk of cardiovascular and pulmonary disease in patients with osteoarthritis.[10 11] Poor sleep can negatively

impact surgical recovery causing slower wound healing, impaired immune function, increased risk of infection and longer hospital stays.[12–15] Surgical patients with preoperative sleep disturbances are at greater risk of developing postoperative delirium and surgical complications.[16] Sleep is bidirectionally linked with pain with poor sleep increasing pain sensitivity and inflammatory markers associated with pain.[15 17] Poor sleep before TKR is associated with increased acute and chronic postsurgical pain, increased analgesic use, reduced joint function and range of motion, lower satisfaction and longer inpatient stays.[8 18 19]

Previous trials on sleep and joint replacement have predominantly focused on perioperative and postoperative pharmacological interventions.[20–22] A recent systematic review identified that improved preoperative sleep reduced pain levels and analgesic consumption after TKR.[23] Guidance from the National Institute for Health and Care Excellence (NICE) and European Alliance of Associations for Rheumatology advises avoidance of pharmacological therapy for long-term management of sleep issues and recommends behavioural approaches as first-line treatment.[24 25] Non-pharmacological sleep interventions are potentially more sustainable and cost-effective, with lower risk of side effects.[26] Our recent systematic review identified no randomised controlled trials (RCTs) evaluating a non-pharmacological intervention targeting sleep in patients waiting for TKR.[27]

This study aimed to evaluate the feasibility of conducting an RCT to evaluate the clinical and cost-effectiveness of REST, a non-pharmacological complex sleep intervention for patients undergoing TKR.

## METHODS
### Study design
REST is a two-centred randomised controlled feasibility trial with 1:1 randomisation and an embedded qualitative study. Participants were recruited from two secondary care National Health Service (NHS) hospitals in England and Wales. The trial was prospectively registered (ISRCTN14233189). The CONSORT checklist is provided in online supplemental appendix 1.

### Patient and public involvement
The study was developed in collaboration with a musculoskeletal patient and public involvement group working in partnership with the University of Bristol and North Bristol NHS Trust. The study benefited from the active involvement and contributions from a group of experienced patient partners. The group comprised five patients with lived experience of knee replacement. They met four times to codesign patient-facing study materials, monitor study progress, provide input into study process, and review the study results and dissemination plans. A patient partner was a member of the Steering committee.

### Participant identification, recruitment and randomisation
Patients waiting for a primary TKR for osteoarthritis were identified from surgical waiting lists. Those likely to have surgery within 3 months were sent a prenotification card by post, followed by a screening questionnaire. The 3-month preoperative time point was selected to allow sufficient time for intervention delivery and engagement to affect behaviour change.

Eligibility criteria were as follows: adults on the TKR waiting list, experiencing disturbed sleep (defined as a score of 0–28 on the Sleep Condition Indicator (SCI) questionnaire, a validated screening tool for insomnia[28]) and access to a device with internet connection. Exclusion criteria were as follows: diagnosed with or receiving treatment for a clinical sleep disorder, taking prescription medication to help with sleep, having taken part in an interventional sleep study in the past 6 months and unable or unwilling to attend an intervention appointment, provide informed consent or complete questionnaires in English. Patients who returned an eligible screening questionnaire were telephoned for an eligibility assessment. Eligible patients were invited to a recruitment visit. Patients not eligible at screening were sent a thank you letter.

After written informed consent was provided and the baseline questionnaire completed, participants were randomly allocated to the intervention plus usual care or usual care alone. Randomisation was conducted on a 1:1 intervention:control basis by the co-ordinating centre using computer generated randomisation. Participants, practitioners and research staff were not blinded.

### Intervention
The REST intervention was developed following Medical Research Council guidance for complex intervention development.[29 30] The Template for Intervention Description and Replication(TIDieR) checklist is provided in online supplemental appendix 2. REST consists of an appointment with an extended scope practitioner (ESP) delivered via videoconference or telephone 3 months before surgery. The 1-hour appointment comprised a comprehensive sleep assessment to identify individual sleep issues and needs, and an assessment of sleep apnoea risk. Participants scoring high risk for sleep apnoea were referred to their General Practitioner (GP) in addition to the intervention. Participants were then provided with tailored sleep education and sleep hygiene advice. One of three existing evidence-based sleep interventions (ESIs) was recommended through a shared decision-making process: cognitive–behavioural therapy for insomnia (delivered via online platform Sleepstation), relaxation (delivered via the Calm app, workbook or guided audio/video) and mindfulness (delivered via the Headspace app, workbook or guided audio/video).

Participants were provided with a personalised sleep plan, which included Specific, Measurable, Achievable, Relevant and Time-bound (SMART) goals based on the sleep hygiene recommendations (eg, reducing coffee

intake, removing electronics from the bedroom, starting a bedtime routine/sleep schedule), a detailed overview of their chosen ESI, instructions for use and digital access (if applicable) and any materials. Participants received a follow-up telephone call 4 weeks after the appointment to review progress and engagement with their sleep plan, calls lasted approximately 30–45 min. This included addressing any barriers experienced, review of the sleep goals and adjustments to the sleep plan if needed.

SMART goals based on the sleep hygiene recommendations (eg, reducing coffee intake, removing electronics from the bedroom, starting a bedtime routine/ sleep schedule), a detailed overview of their chosen ESI, instructions for use and digital access (if applicable) and any materials. Participants received a follow-up telephone call 4 weeks after the appointment to review progress and engagement with their sleep plan, calls lasted approximately 30–45 min. This included addressing any barriers experienced, review of the sleep goals and adjustments to the sleep plan if needed.

All participants received usual care as provided by the participating NHS hospitals. Safety reporting was exclusively for adverse reactions directly attributable to the intervention.

### Intervention delivery training

All practitioners took part in a 1-day online intervention delivery training session. This covered the study background, evidence on the relationship between sleep and pain, an overview of each ESI and practical guidance on delivery. The chief investigator communicated regularly with practitioners to provide further support and training if required. ESPs were provided with a detailed intervention manual, which provided guidance and proformas for conducting the sleep assessment, sleep hygiene and education advice, information on each ESI, participant sleep plan and postappointment tasks.

### Intervention timing

REST was designed to be delivered 3 months presurgery. This time point was chosen to optimise the effect of the sleep interventions, because of the duration of the sleep interventions (Sleepstation is delivered over 6–8 weeks) and theories of behaviour change maintenance.[31]

### Intervention delivery fidelity

Non-participatory observations were conducted to assess the degree to which the intervention was delivered as intended as per the intervention manual. One clinic appointment and follow-up call were observed for each ESP. Observations were conducted independently by two members of the research team. Participants were asked to provide verbal consent for the researcher to be present during their clinic appointment.

### Feasibility outcomes

Feasibility outcomes included recruitment rate, intervention uptake and adherence, outcome data completion,

and intervention acceptability.[32] A full list of outcomes and measurements are outlined in table 1.

### Patient-reported outcomes

Patient-reported outcomes were assessed using paper questionnaires prior to randomisation (approximately 3 months preoperative), 1 week prior to surgery and 3 months after surgery. Participants who did not have their operation by 6 months postrandomisation completed presurgery outcomes.

Outcomes included joint pain (Oxford Knee Score (OKS)[33], neuropathic pain (painDETECT[34]), sleep quality and beliefs about sleep (SCI[28]), Pittsburgh Sleep Quality Index (PSQI)[35]), mental well-being (Hospital Anxiety and Depression Scale[36]) and general health and well-being (EQ-5D-5L,[37] ICEpop CAPability measure for Adults (ICECAP-A)[38]). Health resource use data included healthcare interactions in the community and secondary care, including medication use and were collected in the 3-month preoperative and postoperative questionnaires only. Intervention participants completed sleep treatment engagement questions at the 3 month postsurgery time point.

### Qualitative study

Embedded qualitative work explored the acceptability of the intervention and study processes. Interviews were conducted with participants in both arms and with ESPs delivering the intervention. Participants who expressed an interest at enrolment in being interviewed were sent an invitation letter, reply slip and prepaid envelope. Participants who returned the reply slip were contacted a researcher to discuss participation and arrange an interview for those interested. Informed consent was provided by participants before interview. ESPs were invited to participate in two interviews: one following the intervention training day and one after delivery of intervention appointments. Informed written or recorded verbal consent was provided by all ESPs.

### Qualitative data collection

All interviews were conducted via videoconference or telephone depending on preference. Face-to-face interviews were not possible due to COVID-19 restrictions. Participant interviews were guided by semistructured topic guides (online supplemental appendix 3) covering design and conduct of the trial (all participants), experiences of the intervention and views on impact (intervention group) and changes to made to sleep (all participants). ESP interviews at both time points explored acceptability of training and intervention delivery.

### Progression criteria

Progression criteria for demonstrating the feasibility of an RCT were proposed as ≥60 patients randomised (75% of target) and 75% uptake of the intervention. Uptake was defined as the number of participants who attended an intervention appointment. Criteria for progression based on acceptability were as follows:

**Table 1** Feasibility outcomes

| Objective(s) | Outcome | Measurement |
|---|---|---|
| 1 | Eligibility and recruitment rates | Number of patients invited, returning screening questionnaires, eligible, consented and randomised. Retention rates. |
| 2 | Intervention uptake | Number of participants who attend the clinic appointment |
| 3 | Intervention adherence | Open ended questions at 4-week follow-up telephone call:<br>► Changes made as a result of sleep hygiene and education advice<br>► Engagement in the assigned sleep intervention<br>► Any additional changes made to sleep or sleep routine |
| 4, 6 | Participant interviews: acceptability of the intervention and randomisation | Semistructured qualitative interviews with participants in the intervention group (n=20, 10 per site) and the control group (n=5). |
| 4, 8 | Extended scope practitioner interviews: acceptability of the intervention and training optimisation | Semistructured qualitative interviews with ESPs (n=4) at two time points: (1) after completion of training and (2) after intervention delivery |
| 5 | Intervention delivery fidelity | Observation of one clinic appointment and follow-up call for each ESP to assess adherence and compliance. |
| 7 | Health economics data | Quality of life measures (EQ-5D-5L, ICECAP-A) and healthcare resource use (community and secondary care) as documented in the patient completed outcome measure booklets at 1-week preoperative and 3 months postoperative. |
| 8 | Optimisation of intervention training | Non-participatory observations of the ESP training.<br>Semistructured interviews with all ESPs (n=4) at two time points, after completion of training and after delivering the intervention. |
| 9 | Inform the primary outcome measure for a full trial | Quantitative data analysis, proportion of participants in ongoing pain in each treatment arm at 3 months after surgery. |

ESP, extended scope practitioner.

► All participants: expressed comfort with study processes including recruitment, randomisation, outcome measures and follow-up.
► Intervention participants: level of engagement with clinic processes, adherence to and engagement with the intervention.

### Sample size
The target sample was 80 participants (40 intervention, 40 usual care) to estimate 75% randomisation rate (RCT progression criteria) with 95% CI from 65% to 85%, and to estimate 75% intervention uptake with 95% CI from 60% to 90%.

### Statistical analyses
Baseline characteristics of each group were tabulated using means and SD for normally distributed data, medians and IQRs for non-normally distributed data, and percentages and counts for categorical data. Patient-reported outcome measures were summarised descriptively. The proportion of people without complete responses for the outcome questionnaires were reported at each time point with commentary on any patterns of missing data between time points. Outcome data tables are provided in online supplemental appendix 4.

### Economic analyses
The feasibility of collecting data for an economic evaluation alongside a full trial was assessed, including intervention costs for appointments and tailored intervention. Economic analysis tables are provided in online supplemental file 1.

### Qualitative analysis
Data collection and analysis were conducted in parallel after the first three interviews. Audio files were transcribed, then transcripts were anonymised and imported into the qualitative software package NVivo V.10. Participant and ESP data were analysed separately using framework analysis, a thematic approach that enables structured comparison and contrast of data across cases.[39] Data were organised using the topic guide as a starting framework. Five transcripts were independently double coded and discussed within the team to offer further insight into interpretation and to enhance rigour through different approaches and knowledge.[40] All participants were assigned pseudonyms to ensure anonymity. Participant demographics and supporting quotes are included in online supplemental appendices 5 and 6.

### RESULTS
### Eligibility and recruitment rates
Between March and December 2021, 378 patients were invited to take part in screening. Of these, 258 (68%) returned completed screening questionnaires: 146 were willing to take part and met screening eligibility criteria,

58 declined to take part and 54 were not eligible. Reasons for ineligibility included SCI score ≥29 (28/54, 52%), having a sleep disorder or taking medication to help with sleep (18/54, 33%), operation dates allocated in the near future or surgery being postponed (4/54, 7%), questionnaire returned after study closure (3/54, 6%) and no internet access (1/54, 2%).

Telephone calls were made to the 146 patients who returned an eligible screening questionnaire: 8 were not contactable, 32 were ineligible, 47 declined to take part and 59 consented to take part. Reasons for ineligibility at telephone screening included operation dates allocated in the near future or surgery being postponed (13/32, 41%), having a sleep disorder or taking medication to help with sleep (9/32, 28%), no internet access (7/32, 22%), already had surgery (2/32, 6%) and did not speak English (1/32, 3%). Most did not give a reason for declining taking part. Where given, common reasons for declining were time commitments or personal circumstances (17/47, 36%) and did not feel they had a sleep problem or did not think treatment would help (6/47, 13%).

Two participants withdrew prior to randomisation; therefore 57 participants were randomised.

A Consolidated Standards of Reporting Trials (CONSORT) diagram outlining participant flow is provided in figure 1. Baseline participant characteristics are provided in table 2. Patients who were eligible but did not participate had higher (better) preoperative sleep as measured using the SCI than randomised participants (online supplemental appendices).

### Attendance at the intervention clinic appointment
Of the 28 participants assigned to the intervention group, 3 withdrew before receiving the intervention due to operation dates being allocated in the near future (10.7%), 25 attended the clinic appointment (89.3%) and 15 completed the 4-week follow-up call (53.6%).

### Engagement with the REST intervention and adherence to the agreed sleep plan
#### Qualitative study: participant interviews
38 trial participants were invited to take part in an interview (N=18 Bristol/20 Cardiff; 22 intervention, 16 usual care). 16 expressions of interest were received, and 13 participants were interviewed (N=10 Bristol/3 Cardiff; N=8 intervention/5 usual care). Ethnicity was reported as white British (n=7), English (3), Welsh (n=2) and white (n=1). Marital status was reported as married/partner (n=8), widowed (n=2), single (n=2) and divorced (n=1). Further participant characteristics are described in online supplemental appendix 5.

Participants were willing to discuss their sleep issues with the ESPs. Factors influencing readiness to engage in a conversation with the practitioner included a belief that they were meeting with a knowledgeable and skilled professional. The practitioner's manner and communication style helped to create a safe space that enabled participants to feel both at ease and comfortable to open up about their experiences. Participants who recalled the shared decision-making process of choosing a sleep intervention said that they felt involved and informed.

### Intervention acceptability
#### Patient acceptability
COVID-19 restrictions at the time of delivery required all recruitment and intervention appointments to be conducted remotely by videoconference or telephone. Participants understood the need for remote appointments, but confidence and familiarity with this approach varied. For many participants confidence in using video calls had grown during the pandemic. One participant struggled with confidence and familiarity with online appointments and needed additional support. Remote delivery was generally seen as acceptable. Participants highlighted benefits of removed travel and cost, and reduced risk of COVID-19 and other infections.

Appointment length and structure were considered appropriate. Participants felt they had enough time to ask questions, discuss what was being asked of them and to address concerns. The time to attend the appointment was seen as worthwhile, as it gave access a practitioner who provided the chance to talk about their problems and a focus during their wait for surgery.

Due to clinic delays and competing demands of the practitioners, some appointments started later than planned. Although some participants were accepting of this, one participant reported feeling frustrated and angry at the inconvenience caused.

No adverse events were reported.

#### Practitioner acceptability
ESP acceptability of intervention delivery was high. ESPs were able to deliver most appointments using videoconference which supported better communication with participants, however, one ESP expressed a preference for telephone appointments as this required less set up and had greater flexibility. Some technical issues were raised due to ESPs using different NHS computers for appointments depending on their schedule. This caused problems with webcam connectivity and added additional time to appointment set up. Overall paperwork was straightforward to complete with the questions and proformas clear and easy to use. Some aspects of the assessment were viewed as repetitive and could be shortened to give more time for discussion with participants.

### Intervention delivery
Intervention fidelity assessments included observation of at least one clinic appointment and one follow-up appointment for each practitioner. Practitioners fully or partially met all areas of adherence (fidelity to the intervention as described in the manual) and compliance (proficiency of delivery) during the intervention appointment. Three areas for improvement in training and delivery were identified: educating the participant about sleep and TKR,

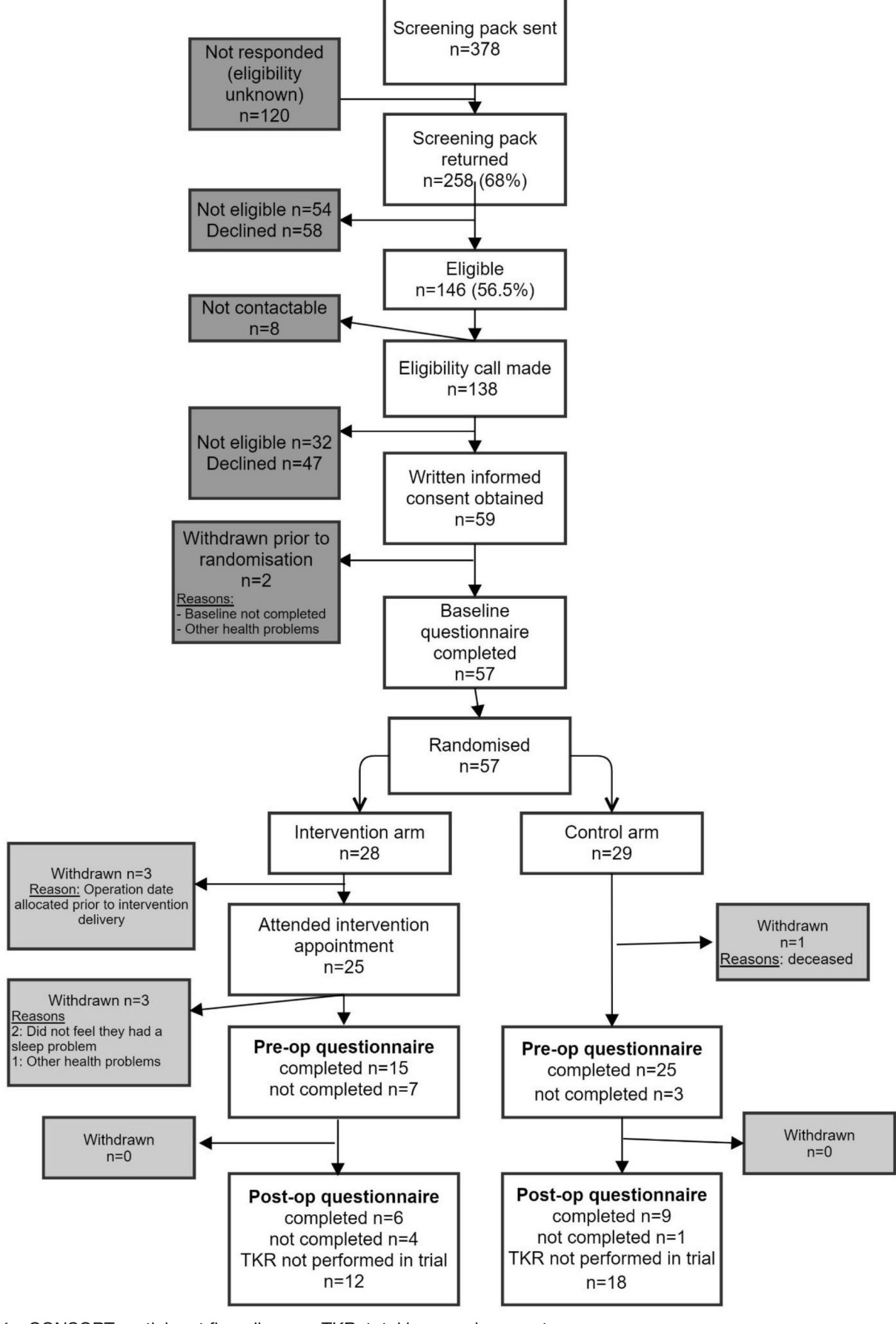

**Figure 1** CONSORT participant flow diagram. TKR, total knee replacement.

**Table 2** Characteristics of study participants at baseline

| Characteristic | N=57* |
|---|---|
| Gender | |
| Man | 23 (40%) |
| Woman | 34 (60%) |
| Ethnicity (cleaned) | |
| Non white/non British | 1 (1.8%) |
| Other/not answered | 4 (7.0%) |
| White/British | 52 (91%) |
| Do you consume any alcohol? | |
| No | 26 (46%) |
| Yes | 30 (54%) |
| Unknown | * |
| Alcohol units/week (excl. non-drinkers) | 9 (3, 14) |
| Non-drinker/unknown | 27 |
| Do you consume any coffee? | |
| No | 16 (28%) |
| Yes | 41 (72%) |
| Cups of coffee/week (excl. non-drinkers) | 12 (7, 20) |
| Non-drinker/unknown | 16 |
| Smoking status | |
| Current | 2 (3.5%) |
| Former | 24 (42%) |
| Never | 31 (54%) |
| Employment status | |
| Employed | 11 (19%) |
| Other | 2 (3.5%) |
| Retired | 44 (77%) |
| Marital status | |
| Divorced | 9 (16%) |
| Married/partner | 36 (63%) |
| Single | 2 (3.5%) |
| Widowed | 10 (18%) |
| Other conditions | |
| No | 18 (34%) |
| Yes (please state) | 35 (66%) |
| Unknown | 4 |
| Other condition(s): other joint replacement | |
| Yes | 4 (100%) |
| Unknown | 53 |
| Other condition(s): pain in other joints | |
| Yes | 4 (100%) |
| Unknown | 53 |
| Other condition(s): arthritis (any) | |
| Yes | 29 (100%) |
| Unknown | 28 |
| Other condition(s): injury | |

Continued

**Table 2** Continued

| Characteristic | N=57* |
|---|---|
| Yes | 3 (100%) |
| Unknown | 54 |

*n (%).

setting SMART sleep goals, and shared-decision-making discussions around recommended sleep interventions.

### Acceptability of randomisation
Qualitative interviews with participants found that most participants found randomisation acceptable and understood the need for this design. Some expressed disappointment on receiving usual care. Disappointment stemmed from their desire to benefit from the intervention because of struggles with sleep.

### Feasibility and acceptability of collecting health economic data
#### Quality of life measures
EQ-5D-5L and ICECAP mean scores and SD with response rates at each of the time point are presented in online supplemental appendices. EQ-5D-5L utility scores were calculated using Hernandez Alava et al's method as recommended by NICE PMG36.[41] No evidence was found of ceiling of floor effects for either quality of life measure, and the measures seem to be responsive to quality of life changes in this population.

#### Resource use
Responses to the bespoke resource use questionnaire were 40/57 (70%) preoperatively and 14/27 (52%) postoperatively. Changes and clarification of questions will be made based on responses to individual questions. For example, more options are needed for physiotherapy appointments as many responses were selected 'other'. Most resource use and cost fell on NHS services including GP, outpatient and physiotherapy appointments. Overall, the estimated cost (NHS perspective, excluding intervention costs) was £507 and £688 for intervention and usual care arms respectively. However, no interpretation of this difference can be made given the small number of participants in this feasibility study.

#### Intervention costs
Given we do not know the value or mechanism of payments from NHS to Sleepstation for use of the app, nor the number of patients purchasing app subscriptions to Calm and Headspace, we costed the intervention based on several assumptions (online supplemental appendices). We have generated a minimum, mean and maximum expected intervention cost. The mean cost of the intervention was estimated at £134.45 per person (£141.04 including patient out-of-pocket costs) based on the mean clinic, preparation and postclinic time captured in the feasibility study, an NHS cost of £147 per patient

for the use of Sleepstation (based on a 50% discount on publicly advertised cost) and one-third of those using Calm or Headspace upgrading to paid subscriptions for additional access. The max estimated at £295.73 and minimum cost £45.29.

## Optimisation of the intervention delivery training package

Four ESPs took part in an interview at time point one (post-training) and three at time point two (postdelivery).

Training was delivered as a 1-day online course. Practitioners spoke positively about their experience of the training, praising the organisation and focus. Practitioners appreciated receiving information about the rationale behind the intervention. Some felt the level of detail around sleep science could be reduced but still found this interesting. They were given sufficient time to ask questions, however, the remote format made this slightly harder compared with face to face. Following experience of intervention delivery, ESPs reported challenges in setting SMART goals including what areas to focus on and the level of detail needed. They suggested more training and knowledge of each ESI would be beneficial and facilitate better shared decision-making discussions with participants. A small number of participants attending appointments felt they did not have a sleep issue. This made it challenging for ESPs to follow the intervention handbook.

Areas for improving practitioner training were identified as:
► Increased time for role-play and practical exercises.
► Additional information on sleep interventions to increase understanding and familiarity.
► Further training and practical exercises on setting SMART goals.
► Advice and guidance on how to support participants who do not believe they have a sleep issue or are not motivated to make changes.
► Additional supervision meetings throughout intervention delivery to provide further support, answer questions and address challenges.

## Data completion rates, selection of the primary outcome measure and sample size for a full trial
### Data completion rates
Data completion rates are provided in table 3. All participants completed the baseline OKS and EQ5D-5L. Completion rates were consistently lower for the PSQI (86%–93%) at baseline and preoperative time points.

| **Table 3** Data completion rates | | | | | | |
|---|---|---|---|---|---|---|
| | **Baseline** | | **Preoperative** | | **Postoperative** | |
| **Characteristic** | **Intervention, N=28*** | **Control, N=29*** | **Intervention, N=15*** | **Control, N=25*** | **Intervention, N=6*** | **Control, N=9*** |
| OKS | | | | | | |
| Complete | 28 (100%) | 29 (100%) | 13 (87%) | 25 (100%) | 6 (100%) | 8 (89%) |
| Missing | | | 2 (13%) | 0 (0%) | 0 (0%) | 1 (11%) |
| PainDETECT | | | | | | |
| Complete | 24 (86%) | 25 (86%) | 9 (60%) | 20 (80%) | 4 (67%) | 6 (67%) |
| Missing | 4 (14%) | 4 (14%) | 6 (40%) | 5 (20%) | 2 (33%) | 3 (33%) |
| SCI | | | | | | |
| Complete | 27 (96%) | 27 (93%) | 15 (100%) | 24 (96%) | 6 (100%) | 9 (100%) |
| Missing | 1 (3.6%) | 2 (6.9%) | 0 (0%) | 1 (4.0%) | | |
| HADS | | | | | | |
| Complete | 27 (96%) | 29 (100%) | 12 (80%) | 25 (100%) | 6 (100%) | 9 (100%) |
| Missing | 1 (3.6%) | 0 (0%) | 3 (20%) | 0 (0%) | | |
| PSQI | | | | | | |
| Complete | 24 (86%) | 27 (93%) | 13 (87%) | 21 (84%) | 6 (100%) | 9 (100%) |
| Missing | 4 (14%) | 2 (6.9%) | 2 (13%) | 4 (16%) | | |
| EQ5D-5L | | | | | | |
| Complete | 28 (100%) | 29 (100%) | 15 (100%) | 25 (100%) | 6 (100%) | 8 (89%) |
| Missing | | | | | 0 (0%) | 1 (11%) |
| ICECAP | | | | | | |
| Complete | 25 (89%) | 29 (100%) | 15 (100%) | 25 (100%) | 6 (100%) | 9 (100%) |
| Missing | 3 (11%) | 0 (0%) | | | | |

Qualitative interviews demonstrated that questionnaire completion was acceptable overall.
*n (%).
HADS, Hospital Anxiety and Depression Scale; OKS, Oxford Knee Score; PSQI, Pittsburgh Sleep Quality Index; SCI, Sleep Condition Indicator.

The painDETECT questionnaire had the lowest completion rates at each time point, ranging from 60% to 86%.

## Patient-reported outcome measures

Baseline, preoperative and postoperative outcome measures are presented table 4. The purpose of this study was to evaluate the feasibility of conducting an RCT, therefore, statistical tests to compare outcomes between treatment arms were not performed.

Participants randomised to the intervention group reported an improvement in average PSQI score from 12.0 (95% CI 8.8 to 14.2) at baseline to 8.0 (95% CI 6 to 11) at the end of the intervention (12 weeks after randomisation), compared with no change in the usual care group (baseline score of 11 (95% CI 8 to 13.5) and 12-week score of 11 (95% CI 7 to 13)).

A proposed primary outcome was pain after surgery as measured by the OKS pain component.[42] The target timing for randomisation was 3 months preoperative. The mean number of days from randomisation to operation was 118 days, with 35% (n=20) of participants having surgery during the 6-month participation window. Outside of the participation window, a further 18 participants were allocated an operation date. The remaining 21 had not been allocated a surgery date at study closure.

## DISCUSSION

This feasibility study has demonstrated that the REST intervention is acceptable to patients and clinicians. With modifications, a full trial is feasible. Criteria for progression to a full trial are ≥60 patients randomised (75% of target) and 75% uptake of the intervention. More than 75% of participants allocated the intervention attended the clinic appointment (89%, n=25/29). We randomised 57 patients during a period of COVID-19 restrictions when many studies were unable to recruit. In addition, removing the need to screen patients who are 3 months prior to surgery would facilitate increased recruitment.

## Strengths

Despite COVID-19 restrictions at the time of the study, screening and recruitment procedures were successful and 57 participants were randomised. Close working with waiting list staff and surgeons was essential to understanding which patients were most likely to be allocated a surgery date in 3 months. Once identified, the return rate for screening questionnaires was 68%. Evidence-based methods to increase the return of postal questionnaires were used, including prenotification cards and non-monetary incentives (individually wrapped tea bags).

Most participants randomised to the intervention group attended an appointment (89%, n=25/29) and engaged with treatment. Remote delivery of the intervention was viewed positively by participants. Those who had a video call who appreciated being able to see the practitioner, welcoming the human connection and chance to build rapport.

## Limitations

Although intervention uptake and engagement with treatment was good, inequalities in access to the internet and electronic devices are an issue in studies that use remote delivery. Some participants experienced delays in obtaining appointment times, which varied by practitioner and site; this may be solved by centralised intervention delivery and offering options such as telephone delivery for those without internet access.

Patients who were eligible at screening but who chose not to take part in the study had better preoperative sleep as measured using the SCI than randomised participants. A common reason for not taking part was not having a sleep problem or feeling that treatment would not help. This indicates the SCI eligibility score cut-off would benefit from being lowered.

Study delivery was redesigned to be conducted entirely remotely to meet COVID-19 restrictions, which also influenced the volume of knee replacement operations performed, affecting the number of participants undergoing surgery within the study.

Generally, NHS operation dates cannot be reliably predicted 3 months in advance, therefore, identifying patients at this time point proved challenging. Completion of the primary outcome at 3 months postsurgery was also difficult because many participants did not have their operation within the study timelines. In addition, variations in length of time from randomisation to 3 months postsurgery would result in high heterogeneity. To address this, the primary outcome assessment for a full trial should not be the proposed outcome of pain 3 months postoperative, but sleep quality at 14 weeks postrandomisation time point for generalisability.

## Modifications

There are several key areas to adapt and improve for a future full trial. These include changes to the clinician training programme, including more detailed training on existing sleep interventions and setting SMART goals, streamlined delivery of the intervention by provision of an online portal, and lowering to the screening cut-off for the SCI score.

A review of equality, diversity and inclusion strategies will ensure a full trial supports inclusivity and engagement from a wide range of communities.

## Conclusions

We have demonstrated that a full RCT is feasible based on the predefined progression criteria and have identified areas for improvement to optimise trial design. Recruitment is achievable, engagement with and adherence to the intervention is high and, importantly, the intervention is acceptable to patients and clinicians.

**Acknowledgements** We would like to thank our patient partners for their help and input, and the research team members and health care professionals who delivered the study including: Gemma Munkenbeck, Leigh Morrison, Paolo Buscemi, Christine Hobson, Giles Head, Lucy Young, Jessica Falatoori, Mat Williams, Chris Dobson and

**Table 4** Patient-reported outcome measures

| | Baseline | | Pre-operative | | Post-operative | |
|---|---|---|---|---|---|---|
| | Intervention, N=28* | Usual care, N=29* | Intervention, N=15* | Usual care, N=25* | Intervention, N=6* | Usual care, N=9* |
| OKS pain subscale | 8.5 (6.0, 10.0) | 9.0 (6.0, 11.0) | 7.0 (6.0, 10.0) | 8.0 (5.0, 12.0) | 13.50 (11.50, 15.50) | 14.50 (12.50, 16.00) |
| OKS function subscale | 7.5 (3.8, 12.0) | 9.0 (7.0, 12.0) | 7.0 (5.0, 9.8) | 8.0 (5.0, 11.0) | 21.5 (19.5, 22.8) | 22.5 (21.0, 24.0) |
| OKS total score | 16 (12, 21) | 18 (13, 22) | 15 (11, 19) | 16 (9, 22) | 34 (32, 40) | 36 (35, 39) |
| PainDetect score | 18 (11, 21) | 13 (9, 19) | 11 (10, 17) | 14 (10, 24) | 7 (6, 12) | 10 (6, 16) |
| PainDetect score (categorised) | | | | | | |
| Ambiguous | 6 (25%) | 5 (20%) | 3 (33%) | 5 (25%) | 0 (0%) | 1 (17%) |
| Neuropathic likely | 9 (38%) | 8 (32%) | 1 (11%) | 7 (35%) | 1 (25%) | 1 (17%) |
| Nociceptive | 9 (38%) | 12 (48%) | 5 (56%) | 8 (40%) | 3 (75%) | 4 (67%) |
| Sleep conditions indicator | 11 (8, 14) | 13 (10, 16) | 14.0 (11.0, 21.0) | 15.0 (9.5, 18.0) | 22 (16, 24) | 24 (18, 27) |
| HADS score | 13 (10, 22) | 16 (12, 19) | 12.5 (9.0, 19.0) | 14.0 (9.0, 20.0) | 9.0 (4.2, 13.0) | 8.0 (6.0, 14.0) |
| HADS score (categorised) | | | | | | |
| Abnormal | 19 (70%) | 22 (76%) | 7 (58%) | 17 (68%) | 3 (50%) | 3 (33%) |
| Borderline abnormal | 5 (19%) | 5 (17%) | 3 (25%) | 6 (24%) | 0 (0%) | 2 (22%) |
| Normal | 3 (11%) | 2 (6.9%) | 2 (17%) | 2 (8.0%) | 3 (50%) | 4 (44%) |
| ICECAP | 0.85 (0.66, 0.92) | 0.84 (0.70, 0.89) | 0.84 (0.55, 0.90) | 0.84 (0.67, 0.91) | 0.92 (0.87, 0.98) | 0.91 (0.89, 0.92) |
| EQ-5D | 0.33 (0.16, 0.57) | 0.54 (0.30, 0.70) | 0.39 (0.22, 0.54) | 0.45 (0.22, 0.60) | 0.67 (0.60, 0.73) | 0.76 (0.73, 0.77) |
| PSQI global score | 12.0 (8.8, 14.2) | 11.0 (8.0, 13.5) | 8.0 (6.0, 11.0) | 11.0 (7.0, 13.0) | 9.5 (5.2, 11.5) | 7.0 (5.0, 9.0) |

*Median (IQR); n (%).
HADS, Hospital Anxiety and Depression Scale; OKS, Oxford Knee Score; PSQI, Pittsburgh Sleep Quality Index.

Mark Brand. The research team acknowledges the support of the NIHR, through the Clinical Research Network. This work was also supported by the NIHR Biomedical Research Centre at University Hospitals Bristol and Weston NHS Foundation Trust and the University of Bristol.

**Contributors** KW was chief investigator and is the guarantor. KW, RG-H, VW, CP, ABurston, ABlom, DR, NH, SW and JG were coapplicants on the grant application to NIHR. All authors were involved in the design, delivery and interpretation of the study. WB was the Trial Manager. JG was responsible for the economic analysis. CP was responsible for the statistical analysis. EJ and KW were responsible for the qualitative analysis. WB, KW, CP and JG have accessed and/or verified the underlying data. WB drafted the manuscript; all authors revised if for important content and approve the final manuscript.

**Funding** This study is funded by the National Institute for Health and Care Research (NIHR) (Research for Patient Benefit; grant reference number NIHR201036). The research team acknowledges the support of the NIHR, through the Clinical Research Network. This work was also supported by the NIHR Biomedical Research Centre at University Hospitals Bristol and Weston NHS Foundation Trust and the University of Bristol.

**Disclaimer** The views expressed are those of the authors and not necessarily those of the NIHR or the Department of Health and Social Care.

**Competing interests** None declared.

**Patient and public involvement** Patients and/or the public were involved in the design, or conduct, or reporting, or dissemination plans of this research. Refer to the Methods section for further details.

**Patient consent for publication** Not applicable.

**Ethics approval** This study received a favourable opinion from the South West— Cornwall & Plymouth Research Ethics Committee, reference 20/SW/0189 and approval from the Health Research Authority and Health and Care Research Wales (IRAS 289761). Participants gave informed consent to participate in the study before taking part.

**Provenance and peer review** Not commissioned; externally peer reviewed.

**Data availability statement** Data are available on reasonable request. Participants were asked on the consent form if they were willing for their information to be shared anonymously with other researchers to support other research in the future. Anonymised data will be stored on the University of Bristol Research Data Storage Facility (https://data.bris.ac.uk) and will be shared via the University of Bristol Research Data Repository within 6 months of the publication of the study results. Access to the data will be restricted reasonable requests to ensure that data are only made available to bona fide researchers after a data access agreement has been signed by an institutional signatory.

**ORCID iDs**
Wendy Bertram http://orcid.org/0000-0001-8234-2052
Chris Penfold http://orcid.org/0000-0001-8654-353X
Vikki Wylde http://orcid.org/0000-0002-8460-1529
Rachael Gooberman-Hill http://orcid.org/0000-0003-3353-2882
Katie Whale http://orcid.org/0000-0002-0012-7103

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
