## [Reviewer comments · BMJ Open]

ARTICLE DETAILS

TITLE (PROVISIONAL)	REST: A pre-operative tailored sleep intervention for patients undergoing total knee replacement: feasibility study for a randomised controlled trial
AUTHORS	Bertram, Wendy; Penfold, Chris; Glynn, Joel; Johnson, Emma; Burston, Amanda; Rayment, Dane; Howells, Nicholas; White, Simon; Wylde, Vikki; Gooberman-Hill, Rachael; Blom, Ashley; Whale, Katie

VERSION 1 – REVIEW

REVIEWER	Frimpong, Emmanuel University of the Witwatersrand Johannesburg
REVIEW RETURNED	25-Sep-2023

GENERAL COMMENTS	GENERAL COMMENT The authors report preliminary findings on a feasibility study of a randomised controlled trial of pre-operative tailored sleep intervention for patients with osteoarthritis undergoing total knee replacement surgery. The authors report promising preliminary findings for designing the actual intervention. However, I have comments or questions to be considered by the authors appended at the various sections of the manuscript. ABSTRACT Briefly highlight the main intervention in the abstract. Article summary Line 46: "COVID" to "COVID-19". INTRODUCTION While the authors included an indication for TKR, this section must be developed further. Authors should briefly highlight the treatment of osteoarthritis and why TKR is important. What is the cost-effectiveness of TKR or surgical success in terms of functional improvements and/or complications following TKR? Please write "NICE" and "EULAR" in full the first time of their usage and then use the abbreviations thereafter. Overall, the significance and clinical relevance of the study is not clear and should be clarified.
---

	What is the “REST” intervention? What are the components of the REST intervention? How is the intervention assessed? These must be thoroughly described in the introduction. The actual intervention of the study has not been introduced. METHODS Eligibility criteria: were other sleep questionnaires used for the sleep problems assessment? How sensitive is the Sleep Condition Indicator (SCI) questionnaire to identifying sleep problems? More relevant details on the REST intervention must be provided – for example; what was the duration per the online session following the one-day online meeting? And how many times per week was the intervention? Were the participants randomised into one of the ESIs? It appears that was not the case. It’s important that authors provide a detailed description of the randomisation. With the current design, how do the authors disentangle the individual or combined effects of the ESIs in this study? The patient-reported outcome questionnaires should be briefly described. Feasibility: what are the justifications for the feasibility criteria chosen? Sample size: how was the sample size of 80 estimated? Statistical analysis: what were the primary and/or secondary outcomes? did the authors analysed changes in the patient-reported outcome measures including the sleep measures? The quantitative analysis is not complete. Were effects sizes calculated? RESULTS AND DISCUSSION The discussion is very scanty and offers no perspective to the very limited descriptive results.
--	---

REVIEWER	Abdelgadir, Ibtihal Sidra Medical and Research Center, Doha , pediatrics
REVIEW RETURNED	03-Oct-2023

GENERAL COMMENTS	This manuscript is well written, however the following points to be considered for revision:  1. Randomisation process is clear, however no mention to allocation concealment plan done. 2. Sample size calculation was done, aim was to recruit 80 participants to achieve the desired outcome results, however the total recruitment was much less than that plan, would that mean no full RCT is recommended? 3. The abstract missed some important points, in both methods and results, would expect PICO summary there, with a summary of the result. 4. Outcomes were shared mainly at the table at the end of the study, and was difficult to follow the narrative discussion within the
--

	text. I would suggest to have the PICO question fully discussed with the results achieved within the text.
--	--

VERSION 1 – AUTHOR RESPONSE

Reviewer 1	
R1: Abstract	
1. Briefly highlight the main intervention in the abstract.	The abstract has been amended to clarify the intervention.
2. Article summary Line 46: “COVID” to “COVID-19”.	This has been amended [line 59].

R1: Introduction	
3. While the authors included an indication for TKR, this section must be developed further. Authors should briefly highlight the treatment of osteoarthritis and why TKR is important. What is the cost-effectiveness of TKR or surgical success in terms of functional improvements and/or complications following TKR?	Thank you for your helpful suggestion. We have expanded on the first paragraph of the introduction to include detail on the need for TKR and expected outcomes for surgical success [lines 73-75].
4. Please write “NICE” and “EULAR” in full the first time of their usage and then use the abbreviations thereafter.	Thank you for identifying this omission. NICE and EULAR have been written in full where they first are used [lines 92 – 93].
5. Overall, the significance and clinical relevance of the study is not clear and should be clarified.	The overall outcome of the study is that it is feasible to test the intervention in a full randomised controlled trial. We have added further information to the introduction to highlight the importance of sleep for TKR patients and the clinical implications of poor sleep [lines 84-86].

6. What is the “REST” intervention? What are the components of the REST intervention? How

Thank you, we have further expanded on our description of the REST intervention to ensure

is the intervention assessed? These must be thoroughly described in the introduction. The actual intervention of the study has not been introduced.

that it is clearly described. In line with CONSORT guidelines, we have included this description in the methods section rather than the introduction [lines 139 – 155].

We have also included a TIDieR checklist [line 141] in the appendices.

R1: Methods

7. Eligibility criteria: were other sleep questionnaires used for the sleep problems assessment? How sensitive is the Sleep Condition Indicator (SCI) questionnaire to identifying sleep problems?

The Sleep Condition Indicator (SCI) was the primary measure used for the sleep problems assessment.

This screening tool is validated to screen for insomnia in the general population and is closely related in validity to the Pittsburgh Sleep Quality Index (PSQI). The PQSI was also collected in both pre- and post-operative study questionnaire. We have added content to the methods section so that this information is clear [line 126].

8. More relevant details on the REST intervention must be provided – for example; what was the duration per the online session following the one-day online meeting? And how

Further details on the REST intervention have been added to the intervention section, starting from line 139. Additions clarify the duration of the intervention appointment with an ESP (one

many times per week was the intervention?

hour), and follow-up appointment (30-45 minutes). Each participant received a personalised sleep plan with SMART goals tailored to their needs, and signposting to one of three existing sleep interventions (ESI). Number of times per week of engagement with sleep hygiene and ESI varied according to the personalised plan [lines 151 – 155].

9. Were the participants randomised into one of the ESIs? It appears that was not the case. It's important that authors provide a detailed description of the randomisation. With the current design, how do the authors disentangle the individual or combined effects of the ESIs in this study?

You are correct, there was no randomisation between ESIs. REST is a complex tailored intervention with sleep hygiene and ESI recommendations based on assessment and shared decision-making. Participants were randomised to either REST or usual care. Tailoring is an important part of REST and supports motivation and behaviour change. In addition, our intervention development work

	indicated a strong preference for tailoring and patient choice. As this was a feasibility study, the effects of the intervention were not examined. The focus was on the engagement with, adherence to, and acceptability of the sleep intervention to see whether a full trial could be delivered.
10. The patient-reported outcome questionnaires should be briefly described.	The patient-reported outcomes section has been updated to clarify that all outcomes were collected using paper questionnaires. The outcome measures within the questionnaires are described in lines 189 – 194.
11. Feasibility: what are the justifications for the feasibility criteria chosen?	The feasibility aims were developed to answer the question of whether a full trial would work, based on guidance published by Orsman and Cohn [line 182]. Orsmond, G. & Cohn, E.S. (2015). The distinctive features of a feasibility study: Objectives and Guiding Questions, Occupational Therapy Journal of Research, 35(3). https://doi.org/10.1177/1539449215578649
12. Sample size: how was the sample size of 80 estimated?	The sample size for the feasibility study was set at 80 participants (40/arm) to estimate 75% randomisation rate (RCT progression criteria) with 95% confidence interval from 65% to 85%, and to estimate 75% intervention uptake with 95% confidence interval from 60% to 90%. This is detailed in lines 221-223.

13. Statistical analysis: what were the primary and/or secondary outcomes? did the authors analysed changes in the patient-reported outcome measures including the sleep measures? The quantitative analysis is not complete. Were effects sizes calculated?

The study design is a feasibility trial and therefore we did not conduct formal statistical comparisons of outcomes between the intervention and usual care group, as recommended in the published literature.

Sim, J. Should treatment effects be estimated in pilot and feasibility studies?. Pilot Feasibility Stud 5, 107 (2019).

<https://doi.org/10.1186/s40814-019-0493-7>

The primary aim of the study was to answer the question of whether a full-scale trial could be done. The full list of feasibility outcomes is provided in Table 1 and to ensure this information is clearer to readers we have amended the abstract so that the primary aim

	of feasibility is provided at the start of the outcomes section.
--	--

R1: Results and discussion	
14. The discussion is very scanty and offers no perspective to the very limited descriptive results.	The aim of this study was to assess if a full randomised control trial of the REST intervention was feasible. We have provided a full and transparent description of our results as they pertain to our aims of evaluating the feasibility of a randomised trial. The discussion section reflects on our findings, considers the strength and limitations of the study and the modifications needed for a full trial. We believe this is an appropriate discussion of the results of this feasibility study.

Reviewer 2	
1. This manuscript is well written, however the following points to be considered for revision:	Thank you for your kind remarks. We have addressed your points below.
2. Sample size calculation was done, aim was to recruit 80 participants to achieve the desired outcome results, however the total recruitment was much less than that plan, would that mean no full RCT is recommended?	Progression criteria outlined in lines 212 – 219. This includes the recruitment rate (≥60 patients randomised) as well as the intervention uptake (75% attendance at the intervention appointment) and acceptability of the intervention. The recommendation to proceed to a full RCT was based on all these criteria rather than just the recruitment rate. Intervention uptake and

	acceptability were good. Although the threshold of 60 randomised participants was not met, 57 was considered to be adequate recruitment due to the COVID-19 restrictions in place at the time of study delivery. This is outlined within the discussion, alongside strategies which would increase recruitment in a full trial [lines 460 – 463].
3. The abstract missed some important points, in both methods and results, would expect PICO summary there, with a summary of the result.	Thank you, the abstract has been updated to provide clarity with the addition of headings for design, setting, participants, intervention and outcomes measures. In addition we have clarified that the primary aim of the study was to answer the question of whether a full-scale trial could be done, lines 19 - 55.

4. Outcomes were shared mainly at the table at the end of the study, and was difficult to follow the narrative discussion within the text. I would suggest to have the PICO question fully discussed with the results achieved within the text.

Thank you for this recommendation. As a feasibility study, the aim was to ascertain whether a full trial was feasible and therefore outcomes were collected to determine acceptability and rates of completion, as reported in lines 385 - 388.

Each of the nine feasibility outcomes outlined in Table 1 are discussed within in the results section.

VERSION 2 – REVIEW

REVIEWER	Frimpong, Emmanuel University of the Witwatersrand Johannesburg
REVIEW RETURNED	08-Jan-2024
GENERAL COMMENTS	Overall, the revision has improved the clarity of the manuscript. However, as I previously suggested, the authors need to adequately describe the "REST intervention" in the introduction. The authors need to also clearly specify the primary and secondary outcome measures that were assessed. Lastly, the manuscript needs a more elaborate discussion.

VERSION 2 – AUTHOR RESPONSE

Reviewer comments	
Overall, the revision has improved the clarity of the manuscript. However, as I previously suggested, the authors need to adequately describe the "REST intervention" in the introduction.	Thank you for your suggestion. We have described the REST intervention within the methods section as recommended by CONSORT guidelines for reporting a feasibility study. These describe the information to include in the introduction as the 'scientific background and explanation of rationale for future definitive trial, and reasons for randomised pilot trial' and the 'specific objective or research questions'. The authors wish to ensure this manuscript conforms to the CONSORT guidelines and therefore the description of the intervention can be found in the methods section.

	Should the editors wish us to follow an alternative established set of guidelines, we would be happy to consider this. Schulz K F, Altman D G, Moher D. CONSORT 2010 Statement: updated guidelines for reporting parallel group randomised trials BMJ 2010; 340 :c332 doi:10.1136/bmj.c332 Eldridge S M, Chan C L, Campbell M J, Bond C M, Hopewell S, Thabane L et al. CONSORT 2010 statement: extension to randomised pilot and feasibility trials BMJ 2016; 355 :i5239 doi:10.1136/bmj.i5239
The authors need to also clearly specify the primary and secondary outcome measures that were assessed.	This manuscript reports on the findings of a feasibility study. Feasibility studies are performed to understand information needed to perform a full randomised trial to test for efficacy and they do not have primary outcome measures. Arain, M., Campbell, M.J., Cooper, C.L. et al. What is a pilot or feasibility study? A review of current practice and editorial policy. BMC Med Res Methodol 10, 67 (2010). https://doi.org/10.1186/1471-2288-10-67 For this reason, we are not able to specify primary or secondary outcome measures.
Lastly, the manuscript needs a more elaborate discussion.	The discussion contains a summary of findings, a review of the strengths and limitations, generalisability of findings and modifications required for a future definitive trial. The authors are mindful of the word count of this manuscript and would be grateful if the reviewer would advise whether there is a specific area of

	the discussion which would benefit from elaboration.
--	--